# *In Silico* Study of the Acquired Resistance Caused by the Secondary Mutations of KRAS G12C Protein Using Long Time Molecular Dynamics Simulation and Markov State Model Analysis

**DOI:** 10.3390/ijms232213845

**Published:** 2022-11-10

**Authors:** Gao Tu, Qing Liu, Yue Qiu, Elaine Lai-Han Leung, Xiaojun Yao

**Affiliations:** 1Dr. Neher’s Biophysics Laboratory for Innovative Drug Discovery, State Key Laboratory of Quality Research in Chinese Medicine, Macau Institute for Applied Research in Medicine and Health, Macau University of Science and Technology, Taipa, Macau SAR 999078, China; 2Cancer Center, Faculty of Health Science, University of Macau, Macau SAR 999078, China; 3MOE Frontiers Science Center for Precision Oncology, University of Macau, Macau SAR 999078, China

**Keywords:** KRAS G12C inhibitor, drug resistance, molecular dynamics, Markov state model

## Abstract

Kirsten rat sarcoma viral oncogene homolog (KRAS) is a small GTPase protein which plays an important role in the treatment of KRAS mutant cancers. The FDA-approved AMG510 and MRTX849 (phase III clinical trials) are two potent KRASG12C-selective inhibitors that target KRAS G12C. However, the drug resistance caused by the second-site mutation in KRAS has emerged, and the mechanisms of drug resistance at atom level are still unclear. To clarify the mechanisms of drug resistance, we conducted long time molecular dynamics simulations (75 μs in total) to study the structural and energetic features of KRAS G12C and its four drug resistant variants to inhibitors. The combined binding free energy calculation and protein−ligand interaction fingerprint revealed that these second-site mutations indeed caused KRAS to produce different degrees of resistance to AMG510 and MRTX849. Furthermore, Markov State Models and 2D-free energy landscapes analysis revealed the difference in conformational changes of mutated KRAS bound with and without inhibitors. Furthermore, the comparative analysis of these systems showed that there were differences in their allosteric signal pathways. These findings provide the molecular mechanism of drug resistance, which helps to guide novel KRAS G12C inhibitor design to overcome drug resistance.

## 1. Introduction

The Kirsten rat sarcoma viral oncogene homolog (KRAS) protein belongs to the group of small GTPases, and it acts as a molecular switch that cycles between the active guanosine-5-triphosphate (GTP)-bound state and the inactive guanosine-5-diphosphate (GDP)-bound form [1]. KRAS-mutation in codons 12 can increase the relative amount of the GTP-bound active form that are capable of driving protumorigenic signaling through downstream effector pathways, such as the mitogen-activated protein kinase (MAPK), phosphatidylinositol 3-kinase (PI3K), and the Ral-GEFs pathways [1,2]. Since KRAS as one of the most frequently mutated oncogenes, it is mutated in approximately 25% of all human cancers, participating in more than 90% of pancreatic ductal adenocarcinomas (PDAC), approximately 30–40% of colon cancers (CRC), and 17% of non-small cell lung cancer (NSCLC) [3]. Therefore, the KRAS-mutation protein has become a focus of drug development for the treatment of KRAS mutant cancers [4,5,6,7]. The three-dimensional (3D) structure of KRAS mainly contains four regions including the phosphate binding loop (P-loop, residues 10–17), switch-I (residues 25–40), switch-II (residues 57–76) and nucleobase binding loops (residues 116–120 and 145–147) [8] (Figure 1A).

In recent years, several clinically relevant inhibitors against the KRAS G12C protein have achieved a breakthrough therapy in the treatment of NSCLC [9,10,11,12]. Among them, MRTX849 (adagrasib) has advanced into phase III clinical trials [13] and AMG510 (sotorasib) was approved by the U.S. Food and Drug Administration (FDA) in 2021 (Figure 1B,C) [14]. In spite of the great successes, in vitro experiments and/or in clinical trials show that second-site mutations in KRAS G12C (such as K16T, R68S, Y96C and Y96D) protein could cause different levels of resistance to KRAS G12C inhibitors [15,16,17]. Therefore, to develop potent and effective inhibitors against the double mutant KRAS, the detailed understanding of drug resistance mechanism to small molecule inhibitors against mutant KRAS at the atomic level will be of some urgency.

Up to now, multiple molecular modeling approaches have been proposed to probe the molecular basis of drug resistant mechanisms due to protein residue mutations and ligand binding. The success case include the drug resistance mechanism studies against inhibitors to HIV protease [18,19,20,21], HCV NS3/4A protease [22], and KRAS protein [23,24] etc. MD simulation and binding free energy calculation can provide structural dynamics and mechanisms of interaction between the inhibitors and the targets. Markov State Models (MSM) have shown to be a very useful tool in quantifying the thermodynamics and kinetics of proteins [25]. In addition, dynamic network analysis can offer insights into the key allosteric interaction network in the RAS protein’s structure [26,27].

To elucidate the molecular mechanism of drug resistance of mutated KRAS (G12C-K16T, G12C-R68S, and G12C-Y96C/D) protein toward AMG510 and MRTX849 inhibitor. In this work, based on the available crystal 3D-structures structures of AMG510 and MRTX849 in complex with the KRAS G12C protein [28,29], molecular dynamics (MD) simulation, binding free energy calculation, protein-ligand interaction fingerprint and binding mode analysis were used to explore the drug resistance mechanism of AMG510 and MRTX849 to the mutated KRAS protein. Furthermore, the Markov state model (MSM), 2D-free energy landscapes and dynamical network analysis were also applied to characterize and compare the conformational dynamics of the apo and inhibitor-bound structure of the mutated KRAS complex (G12C-K16T, G12C-R68S, and G12C-Y96C/D). We expect that this work can reveal the acquired resistance mechanisms of KRAS G12C inhibitors and provide the basics of structure-based drug design for overcoming the resistance of novel KRAS G12C inhibitor with improved potency.

## 2. Results and Discussion

### 2.1. Molecular Dynamics Simulations

Starting from the constructed simulated systems, a total of 75 μs (15 × 5000 ns) molecular dynamic simulations were performed in each system, of which ten are AMG510- and MRTX849-bound complexes, and five are apo form KRAS without inhibitors. To monitor the overall stability of each trajectory, the Root Mean Square Deviation (RMSD) values of protein backbone atoms with respect to the initial structure were computed. As shown in Appendix A, the RMSD values of AMG510 bound to G12C, G12C-R68S, G12C-K16T, G12C-Y96C, and G12C-Y96D in the simulation fluctuates around 1.59 Å, 2.55 Å, 2.11 Å, 1.88 Å and 2.06 Å during 5000 ns MD simulation, and this fluctuates around 2.18 Å, 2.23 Å, 2.01 Å, 1.86 Å and 1.89 Å in the complexes of MRTX849 bound G12C, G12C-R68S, G12C-K16T, G12C-Y96C and G12C-Y96D. RMSD analysis indicates that the conformation of four second-site mutation of AMG510- and MRTX849-bound complexes are slightly different from that of inhibitor-KRAS G12C.

### 2.2. Binding Free Energy of AMG510 and MRTX849 to Mutant KRAS

Due to double-site mutations that occur at the inhibitor binding pocket, namely the switch-II pocket. These secondary mutations may directly impact inhibitor binding by altering the binding affinity or modulating the conformational dynamics of protein, which lead to resistance to KRAS G12C inhibitors. To estimate the decreased of binding affinity of inhibitors binding to four double mutant KRAS relative to KRAS G12C protein, the experimental IC_50_ values of the two inhibitors for each system were converted into experimental binding free energies value (ΔG_exp_) by using the ΔG_exp_ = RTln(IC_50_) equation. Furthermore, considering the entropic contributions (−TΔS) upon binding of inhibitors to KRAS were performed for the purpose of predict the ΔG_cal_ more accurately. Table 1 and Table 2 show the covalent binding free energies of AMG510 and MRTX849 in complex with each mutated KRAS protein by the MM/GBSA method using 5000 snapshots taken from the MD trajectories. The correlation coefficients between the calculated (ΔG_cal_) and experimental binding free energies value (ΔG_exp_) of the five AMG510- and MRTX849-bound complexes were 0.70 and 0.95 with a good confidence interval, respectively, which are in a reasonable agreement with the experimental values (Figure 2). In addition, statistically significant differences were observed between inhibitor-bound KRAS G12C and inhibitor bind to each doubled-mutated protein in terms of binding free energy (ΔG_cal_) (Appendix A). In AMG510 bound complexes, the corresponding binding free energies between AMG510 and KRAS G12C, G12C-R68S, G12C-K16T, G12C-Y96C and G12C-Y96D are −31.43, −22.58, −23.54, −25.88 and −21.40 kcal/mol, respectively (Figure 2A). For MRTX840 binding to KRAS G12C, G12C-R68S, G12C-K16T, G12C-Y96C and G12C-Y96D protein, the binding free energy are −38.81, −31.34, −29.26, −30.73 and −28.18 kcal/mol, respectively. (Figure 2B). To better understand the binding mechanisms, comparison of the separate free energy components between the G12C and double mutant KRAS can reveal the origin of drug resistance induced by mutations. Here, both the electrostatic energy (ΔE_elec_) and intermolecular van der Waals energy (ΔE_vdW_) have a favorable contribution to the calculated binding free energy, whereas the polar solvation terms (ΔG_solv_) have an unfavorable contribution to the calculated binding free energy (Table 1 and Table 2). Collectively, our data suggest that the reduced binding affinity between double mutant KRAS and inhibitors are the major factors responsible for drug resistance.

### 2.3. Insight from the Per-Residue Binding Free Energy Calculation

#### 2.3.1. Per-Residue Energy Decomposition for AMG510

Since the binding affinity of inhibitors to the four double mutant KRAS are reduced by different degrees compared with KRAS G12C. An interesting question is which residues lead to the reduction of energy contribution between the protein-ligand complex? To answer this question, the differences between the per-residue energy decomposition of AMG510 and MRTX849 with individual residues in each KRAS-inhibitor complex were calculated (Figure 3). 

Compared to AMG510-KRAS G12C, the second-site mutation residues R68S, K16T, Y96C and Y96D are significantly decreased by 0.56 kcal/mol, 2.08 kcal/mol, 2.19 kcal/mol and 4.27 kcal/mol, respectively (Figure 3A and Appendix A). Except for the four double site mutations on KRAS, other residues such as Q16 and S68 also reduced the binding affinity in AMG510 with G12C-R68S, but both the residues Q61 and S68 of the free energy changes are greater than 0.5 kcal/mol and less than 2.0 kcal/mol. Apart from the G12C-K16T, the change in per-residue absolute energy contribution of residues (G10 and A11) are greater than 0.5 kcal/mol compared to AMG510-KRAS G12C. That is the reason why the absolute binding affinity of AMG510-G12C-K16T is lower than G12C-Y96C. Of all the secondary mutations of KRAS, the Y96D mutation also can significantly reduce the binding affinity of AMG510-KRAS G12C-Y96D almost twice as much as G12C-Y96C. These results indicate that the mutation Y96D changes the hydrophobic nature of the switch-Ⅱ binding pocket, which impairs the molecular interaction with AMG510 [17]. Thus, the negatively charged residue D96 should be fully considered when designing inhibitors for G12C-Y96D. Overall, per-residue energy decomposition analysis indicates that the reduction of energy contribution of secondary mutations is the main reason for the decreased binding affinities of AMG510 and mutated proteins.

#### 2.3.2. Per-Residue Energy-Decomposition for MRTX849

Compared to MRTX849-KRAS G12C, the interaction energy of mutated residues R68S, Y96C and Y96D are significantly decreased by 0.68, 1.46 and 2.82 kcal/mol for MRTX849, except for G12C-K16T (Figure 3B). Furthermore, the interaction energy of residues V9, G10, T58, E62 and M72 in G12C-R68S, G12C-K16T, G12C-Y96C and G12C-Y96D with MRTX84 are simultaneously decreased in comparison with MRTX849-KRAS G12C. To the G12C-K16T system, the K16T mutation is not significantly decreased the per-residue energy contribution of MRTX849 with protein. As shown in Appendix A, the G12C-Y96C/D mutation causes a reduction in the van der Waals energies of Y96C (2.25 kcal/mol) and Y96D (2.12 kcal/mol) interacting with MRTX849 compared to MRTX849-KRAS G12C, which implied that the contribution of van der Waals energy is the key driving force in the binding of MRTX849 to G12C-Y96C. To further reveal the conformational changes caused by double mutations on protein, the dynamical network analysis will be analyzed in Section 3.8.

In summary, the calculated binding free energy of five MRTX849-bound complexes are stronger than that of five AMG510-bound complexes (Table 1 and Table 2). According to the theory of statistical thermodynamics, protein conformational states with higher negative binding energy are more likely to appear in a stable conformation. Thus, the probability of MRTX849 bind to five types of KRAS are higher than that of five AMG510-bound complexes, and the order of probabilities of these two inhibitors binding to five types of KRAS are as follows: KRAS G12C > G12C-R68S > G12C-Y96C > G12C-K16T > G12C-Y96D. We assume that inhibitors have a similar inhibitory effect on the five types of mutated KRAS activities. Then the resistance of the above four secondary mutations to MRTX849 are weaker than that of AMG510-boud complexes, and its drug resistance varies from weak to strong is G12C-R68S < G12C-Y96C < G12C-K16T < G12C-Y96D. Therefore, secondary mutations of KRAS resistance were mainly dominated by inhibitor-bound state based on the above results.

### 2.4. Binding Free Energy of Mutated KRAS G12C and GDP

Although the mutated KRAS resistance to inhibitor was dominated by the inhibitor’s affinity in this study. The correlation coefficients analysis indicated that the drug resistance was related with other factors that account for at least 5% of conformational changes. To explore the influence of the affinity of mutated KRAS and GDP caused by the AMG510 and MRTX849, 5000 frames were extracted from each system to calculate the binding free energy and per-residue energy decomposition (Appendix A and Figure 4). As shown in Appendix A, the predicted ΔG_cal_ values of five apo form mutated KRAS/GDP, five AMG510/GDP and five MRTX849/GDP are −74.72, −60.23, −35.53, −74.29, −77.23, −73.35, −76.72, −40.64, −78.76, −79.65, −64.52, −63.38, −32.64, −61.98, −66.34 kcal/mol, respectively. It is observed that the double mutant KRAS activity was partially restored are attributed to the lower affinity between KRAS and GDP, resulting in the GDP molecule being easily released from the switch-Ⅰ binding pocket. Notably, the K16T mutation significantly reduced the binding affinity of GDP to double-mutated KRAS, and the double mutants G12C-K16T in apo and AMG510/GDP show no anion-cation interaction between residue T16 and GDP as compared to KRAS G12C (Appendix A).

For comparison, the predicted ΔG_cal_ value of apo G12C (−74.72 kcal/mol) is used as a reference value. When the binding energy of GDP is lower than that of the reference value, it is easier to dissociate GDP from the switch-Ⅰ binding pocket, which is helpful for the recruitment of GTP to restore its catalytic activity [30]. On the contrary, it is unfavorable for the inactivated KRAS to convert to its active state. Accordingly, the apo KRAS G12C-R68S and G12C-K16T can be converted to the active state with a higher probability, however the opposite occurs in the case of the apo KRAS G12C-Y96C and G12C-Y96D. Thus, the secondary mutations itself could increase or decrease the affinity of GDP in the switch-I pocket in other ways.

Compared with the systems without inhibitor, AMG510 do not cause a significant change in binding energy between mutated KRAS and GDP, which indicates that the AMG510 binding does not significantly affect the affinity of GDP in KRAS G12C (Figure 4). The double mutants G12C-K16T may promote the release of GDP in the bound state of AMG510, but instead in double mutants G12C-Y96C/D and G12C-R68S compared with binding free energy of apo GDP-KRAS G12C. Compared with the corresponding double mutants of apo KRAS, the calculated binding energies of GDP to the G12C-R68S with AMG510 were significantly decreased by ~20 kcal/mol and increased by ~10 kcal/mol, respectively (Figure 4A), which implies that the affinity of GDP in the two double mutants may have a greater impact with AMG510 binding.

The calculated binding energies of GDP in five MRTX849-bound complexes were lower than ~10.20 kcal/mol compared to the reference value (Figure 4C). This denotes that GDP in the bound state of MRTX849 has a high probability of being released from the switch-Ⅰ pocket, thereby promoting the recruitment of GTP with KRAS to the active state. Compared with the corresponding apo KRAS, GDP in both the single mutant G12C and the three MRTX849-bound states (except for the MRTX849-G12C-K16T) exhibited higher binding energies with an increase of at least 10.0 kcal/mol. The results indicate that the four second mutations in MRTX849-bound complexes may significantly weaken the affinity of GDP, thereby appreciably facilitating the release of GDP from the KRAS protein. Considering KRAS G12C activity inhibition from the GDP point of view, the MRTX849 binding is unfavorable. For the MRX849 in three double mutants (G12C-R68S, G12C-Y96C and G12C-Y96D), GDP release from switch-I was easier than the apo KRAS and then restore the activity of KRAS. Therefore, the reduced affinity of GDP with the above three double mutant MRTX849-KRAS is one of the reasons for the inhibitor resistance. Moreover, Figure 4 also showed that the K16T mutation in each system was predicted to unfavorably decrease the total energy contribution, and the contributions are decreased by 8.88, 12.25 and 14.30 kcal/mol, as compared to the apo form, AMG510- and MRTX849-bound KRAS G12C, respectively. Results reveal that the K16T mutation dominates the affinity change of GDP. The energy contribution loss may come from the side chain of threonine being shorter than that of lysine, resulting in anion-cation interaction frequency decreases between residues T16 and GDP (Appendix A). Therefore, the resistance of KRAS G12C-K16T should be attributed to the reduced affinity of both the inhibitor and GDP.

In conclusion, the reduced affinity of GDP may be one of the reasons for the resistance of MRTX849 to double mutant KRAS, while the molecular mechanism of resistance of AMG510 with double mutated KRAS may not participate in the replacement of GDP to its active state.

### 2.5. Binding Mode between AMG510 and MRTX849 with G12C and Double Mutant KRAS

Figure 5 and Figure 6 depicted the detailed structural comparison of the representative structures for AMG510 and MRTX849 in complex with KRAS G12C and double-mutant KRAS. The representative conformation of each complex is from their conformational ensemble as inferred from the corresponding RMSD using the average structure as a reference of the MD trajectory. All these important residues with a total energy of 89.67% and 85.19% of the residues towards inhibitor binding were located within P-loop (G10, A11 and G13 in AMG510-bound complexes; V9, G10, A11 and G13 in MRTX849-bound complexes), switch-II (Q61, R68, D69 and M72 in AMG510-bound complexes; T58, A59, G60, Q61, E62, Y64 and M72 in MRTX849-bound complexes) and α3-helix (H95, Y96, Q99, I100 and V103 in AMG510-bound complexes; H95, Y96, Q99, I100 and V103 in MRTX849-bound complexes) regions of KRAS. These identified residues based on energy calculations play an essential role in molecular recognition in protein-ligand interactions.

As shown in Figure 5A, AMG510 binds to the KRAS G12C protein mainly through hydrophobic and hydrogen bond contacts. The acrylamide warhead of AMG510 forms hydrogen bond interactions with the main chain of K16, and hydrogen bonds are also between the phenyl ring groups of AMG510 interact with the side chains of R68 and D69. Meanwhile, the moiety of pyridyl ring of AMG510 extends into the cryptic pocket comprised of three residues (H95, Y96 and Q99), which forms a strong hydrophobic interaction. In addition, the side chains of R68 participated in cation-π interactions with AMG510. In the case of MRTX849 bound to KRAS G12C complexes, the binding site of residues are mainly constituted from switch-II (T58, G60, Q61, E62 and Y64) and α3-helix (H95, Y96, Q99, I100 and V103) regions (Figure 6A). As shown in Figure 5C,C1, the K16T mutation also leads to disappearance of hydrogen bonding with AMG510 due to the decrease in the interaction energy is mainly from van der Waals energies contribution (Appendix A), which implied that the key interaction is highly relevant to the reduction of the binding free energy of the inhibitor. The tetrahydropyridopyrimidine core and naphthyl moiety of MRTX849 mainly participated in hydrophobic interaction with cryptic pocket (H95, Y96, and Q99) in the α3 regions of mutant KRAS. Meanwhile, the backbone atom of G10 in KRAS G12C formed hydrogen bond interactions with MRTX849, as well as the side chain of E62 and H95 also formed two hydrogen bonds interaction with inhibitors. From the above analysis, the stability of key interactions of AMG510 and MRTX849 in the switch-Ⅱ binding pocket of KRAS G12C were maintained throughout the MD simulations (Figure 5A and Figure 6A). Moreover, the pharmacophore models of the ten representative structures illustrated that hydrophobic interactions and hydrogen bond interactions are the predominant features for the mutated KRAS protein (Appendix A). For the KRAS G12C complexes, the interactions obtained from the representative structures revealed that inhibitors predominantly formed hydrophobic interactions with the switch-Ⅱ binding pocket (K16, R68, M72, H95, Y96 and Q99) and made hydrogen bonds with residues (K16, R68, H95, E62 and D69) in the KRAS G12C protein. Compared to inhibitor-KRAS G12C complexes, some pharmacophore features have been lost in double-mutated complexes. The results were consistent with the identified binding mode in the switch-Ⅱ binding site for inhibitors. A pharmacophore model and binding mode analysis are helpful in the design of novel KRAS G12C inhibitors to overcome drug resistance.

In Figure 5 and Figure 6, we also find that there are some differences between the conformational features of KRAS G12 and double-mutant KRAS upon AMG510 and MRTX849 binding. Previous studies have shown that two switch regions of KRAS protein play important roles in signal transduction pathways [31]. To describe the dynamics of these functional conformational changes of the protein induced by mutations and inhibitors binding, which will be further discussed in the next section.

### 2.6. Conformational Dynamics Comparison between the Apo Form KRAS and Inhibitor Bind to KRAS Complex

#### 2.6.1. Structural Fluctuation for the Apo and Inhibitor Complex

To further reveal the conformational changes in flexibility of KRAS due to mutations, the root mean squared fluctuations (RMSF) of protein backbone between the KRAS G12C, G12C-K16T, G12C-R68S, G12C-Y96C and G12C-Y96D in the apo form and bound to AMG510 and MRTX849 are calculated (Figure 7A–C). The two loop regions (Loop-I and Loop-II) exhibited highly dynamic behavior demonstrated by RMSF analysis, which revealed major fluctuations in the KRAS in these regions [23]. In the apo form and MRTX849-bound complexes, all systems showed similar fluctuation behaviors. In the AMG510-bound complex, we found that all the double mutated KRAS proteins show very high fluctuations at two loop regions when compared to the KRAS G12C protein (Figure 7B). It is clear from the RMSF result that AMG510 bound to the KRAS G12C protein are more stable than the double mutated KRAS complexes.

#### 2.6.2. Results of MSM

The MSM construction was applied to study the overall dynamic behaviors of the mutated KRAS based on the statistical analysis of MD simulations. Using dimensionality reduction and MSM to study global conformational changes of complex biomolecules is time-consuming and challenging [32]. Thus, to choose an appropriate structural feature to describe the slowest dynamics of these functional con-formational changes is critical. In this study, to describe the functionally relevant two Loop motions, we selected the certain two loop regions residues of 3D coordinates as the feature selection of MSMs due to the high fluctuation (Figure 7). As shown in Appendix A, both the systems of the apo form mutated KRAS and inhibitor-KRAS complex are divided into 10~14 metastates with a variety of proportions. The metastable state (Si) conformations with a total of probability flux > 60.0% were analyzed for each system (Figure 8).

In the case of apo G12C form and inhibitor-bound KRAS G12C protein, the main flux pathway involves two states (state 10 and 12) in G12C, two states (state 5 and 10) in the AMG510-bound complex, and two states (state 6 and 8) in the MRTX849-bound complex. The conformational analysis indicated that the end state 12 in apo G12C appear to reside in the fully opened switch-I conformation, whereas both the end state 10 and 8 in AMG510- and MRTX849-bound complexes appear to reside at the closed conformation (Figure 8A,A1,A2). This comparative conformational analysis between apo and inhibitor-bound KRAS G12C indicated that inhibitors can notably reduce movement in the switch-Ⅰ region, locking the KRAS G12C protein in an inactive state.

For the five apo form KRAS protein, the main flux pathway involves two states (state 10 and 12) in G12C, three states (state 8,13 and 14) in G12C-R68S, two states (state 13 and 14) in G12C-K16T, three states (state 1, 5 and 14) in G12C-Y96C and four states (state 5, 10 and 11) in G12C-Y96D. All the end states in each system primarily reside in the opened switch-I conformation, except for G12C-Y96C protein (Figure 8A–E). These results indicate a mutation in the apo system occurring at the inhibitor binding site can strengthen the flexibility of KRAS and induce more incompact switch-Ⅰ domain in comparison with the initial crystal of KRAS.

With regard to the KRAS G12C, G12C-R68S, G12C-K16T and G12C-Y96C/D bound to AMG510 and MRTX849, the main flux pathway involves two states (state 5 and 10) in G12C, two states (state 8,13 and 14) in G12C-R68S, two states (state 13 and 14) in G12C-K16T, three states (state 1, 5 and 14) in G12C-Y96C and three states (state 5, 10 and 11) in G12C-Y96D with AMG510;the corresponding two states (state 6 and 8) in G12C, two states (state 7 and 14) in G12C-R68S, three states (state 7, 11 and 12) in G12C-K16T, two states (state 10 and 12) in G12C-Y96C and three states (state 9, 11 and 12) in G12C-Y96D with MRTX849 (Figure 8A1–E1,A2–E2). Aside from G12C-R68S and G12C-Y96C with AMG510, the end metastable states in these systems are opened switch-I conformation compared to KRAS G12C complexes (Figure 8A1–E1,A2–E2). The above analysis implies that the K16T, R68S and Y96C/D mutations evidently affect the conformational alterations of the loop regions between the closed and open states. In addition, by alignment of each system with the initial crystal 3D-structure display that second site mutation generates evident influences on conformational alterations of the switch domain while the GDP and Mg^2+^ are aligned well.

#### 2.6.3. Mutation-Mediated Impacts on Free Energy Profiles of Apo and Inhibitor KRAS Complex

To understand the energetic basis with regard to the influences of mutated KRAS, free energy landscapes are constructed by utilizing all MD trajectories on the RMSD values of Loop Ⅰ (residues 29–40) and Loop Ⅱ (residues 57–73) as reaction coordinates. According to the above analysis, RMSDs in the two loop regions of KRAS can efficiently embody conformational transformation of the switch domain of KRAS through the MD trajectory. The higher the RMSD value, the greater the deviation from the crystal structure. By projecting the average RMSD of two loop regions from each metastate onto the 2D-PMF, the constructed free energy profiles (2D PMF) of KRAS G12C and its double-mutant and the representative structures represent different energy basins are detected by MD simulations (Figure 8 and Figure 9). The results found that binding site mutation led to significant changes of free energy landscapes and conformational rearrangement of the mutated apo system and inhibitor-KRAS complex compared to the KRAS G12C.

For the apo form G12C and inhibitor-bound KRAS G12C complexes, MD simulations capture two or three different low-energy wells (Figure 9), indicating that the conformations of the apo form KRAS G12C are mainly distributed in these energetic spaces. According to the 2D-PMF, the RMSD of Loop Ⅰ regions in low-energy wells is significantly smaller than the apo system and MRTX849-KRAS G12C, with ~8.19 Å RMSD on average for apo G12C, ~2.10 Å and ~4.5 Å for AMG510 and MRTX849 bound to KRAS G12C. The most populated metastable state (S10 in AMG510-KRAS G12C; S8 in MRTX849-KRAS G12C) exhibits a closed conformation in Loop 1 regions compared to apo KRAS G12C (Figure 8 and Figure 9). This indicates that the smaller fluctuation of Loop 1 regions may inhibit the KRAS signaling, maintaining inhibitor-KRAS G12C in a GDP-bound inactive state (Figure 8A1,A2).

With regard to the Apo form KRAS G12C and G12C-R68S, G12C-k16T, G12C-Y96C/D, MD simulations identify one or two low-energetic wells, and this result implies that the conformations of the apo form mutated KRAS are primarily populated at distributed in these conformational subspaces (Figure 8). In these low energetic states, the average RMSD of Loop Ⅰ and loop Ⅱ regions between G12C, G12C-R68S, G12C-k16T, G12C-Y96C and G12C-Y96D were located at (~7.50 Å, ~4.52 Å), (~4.75 Å, ~5.00 Å), (~4.25 Å, ~4.75 Å), (~8.00 Å, ~4.50 Å), (~2.20Å, ~4.00 Å), respectively (Figure 9). These results suggests that mutation in the apo system induces more incompact loop-Ⅰ domain by comparison with initial G12C structure and further affects the binding of loop-1 to downstream effector proteins.

In the case of the KRAS G12C, G12C-R68S, G12C-K16T and G12C-Y96C/D bound to AMG510 and MRTX849, one or two energetic low-energetic wells were recognized by MD simulations (Figure 9), signifying that the conformation of the mutated KRAS are populated in theses structural subspaces. The average RMSD of Loop Ⅰ and loop Ⅱ regions in the low energy wells between G12C, G12C-R68S, G12C-k16T, G12C-Y96C and G12C-Y96D are (~2.11 Å, ~3.75 Å), (~2.25 Å, ~5.35 Å), (~4.10 Å, ~4.00 Å), (~2.00 Å, ~3.00 Å), (~4.00 Å, ~3.75 Å) for AMG510, and the corresponding RMSDs are (~4.50 Å, ~3.75 Å), (~6.00 Å, ~5.65 Å), (~3.75 Å, ~3.50 Å), (~2.25 Å, ~4.05 Å), (~3.75 Å, ~3,5 Å) for MRTX849, respectively. By alignment of representative structures falling into the predominant 2D-free energy basins, we found that all MD-conformations of inhibitor-bound KRAS G12C are not located in (0, 0) compared to the initial crystal structure. This implies that the two Loop regions are highly flexible (Figure 9). Hence, it might provide some clues when designing a novel KRAS G12C inhibitor in the future, an improvement in two loop regions of protein-ligand interaction should be considered. Furthermore, the alignment of each representative conformation also indicates that the two loop regions significantly deviate from each other (Appendix A). For the Apo G12C and inhibitor-bound KRAS G12C complexes, the LoopⅠ domains of Apo-G12C, AMG510- and MRTX849-bound KRAS G12C, exhibit open (8.69 Å), closed (1.96 Å) and closed (1.93 Å) conformation compared to initial crystal structure. The results indicate that the inhibitor can reduce the flexibility of Loop Ⅰ regions and mainly maintained at GDP-bound inactive conformation, while the double-mutated complexes with or without inhibitors display different variability compared to KRAS G12C. These variable switch-I configurations may affect the binding of effectors such as SOS and facilitates the exchange of GDP with GTP for KRAS reactivation [23,30,33].

To quantitatively characterize the changes of GDP-bound inactive conformation, we calculated the percentage of two GDP-bound inactive conformation types (type A and type B) in each simulated system. Figure 10 shows that the majority of KRAS G12C protein with AMG150 (type B, 14.41%) and MRTX849 (type A, 23.97%) tend to be in an inactive conformation during the MD simulation. In the AMG510-bound complexes, all the percentage of conformational type A in four double-mutated KRAS are lower than AMG510-KRAS G12C (Figure 10B). However, compared to KRAS G12C bound with MRTX849, no significant change was seen in the double-mutated MRTX849-bound complexes, except for MRTX849-G12C-K16T (Figure 10C). Our results suggest that the higher proportion of two conformational states for AMG510-KRAS G12C are favorable for inhibiting KRAS protein activity. While the conformation type A in each apo state and MRTX849-bound complex do not vary significantly. In particular, the K16T mutation reduces the binding affinity of GDP to each KRAS G12C-K16T system, which may facilitate the release of GDP and then convert it from the inactive state to an active state (Appendix A).

### 2.7. Comparison of Protein-Ligand Interaction Fingerprint between KRAS G12C and Double-Mutant KRAS

To quantitatively determine those interaction differences, the molecular fingerprint of interactions which was calculated with more than 30% of the interaction frequencies. Table 3 and Table 4 and Figure 11 provides a statistical result of the molecular fingerprint. In the five AMG510-bound complexes, twelve residues that interact with AMG510 the most frequently are: V9, A11, T58, A59, G60, Q61, E63, M72, H95, Y96C/D, I100 and V103 which are all in contact with the inhibitor at least 30% of frames. Particularly, the hydrophobic interactions for residues (H95, Y96 and Q99) in KRAS G12C occur respectively in 99.66%, 100.00% and 99.98% during MD simulation (Table 3), while the interaction frequency for residues S68, T16, C96 and D96 in double mutant KRAS interact with AMG510 are decreased by 61.20%, 73.72%, 1.56% and 14.18%, respectively, compared to KRAS G12C. The interaction fingerprints further suggested that the second-site mutation at position 16, 68 and 96 could potentially affect residue-AMG510 interaction and decrease the binding affinity. Besides, it is observed that the interaction frequency of E62 and AMG510 completely disappeared in four double-mutated KRAS.

For the MRTX849-bound complexes, twelve residues interact with MRTX849, and the inhibitor binding site residues are: V9, T58, A59, G60, Q61, R68S, M72, H95, Y96C/D, I100 and V103, which are all in contact with the inhibitor in at least 30% of frames. Particularly, the hydrophobic interaction was mainly observed between residues (V9, M72, H95, Y96 and Q99) in KRAS G12C and MRTX849 with a frequency of more than 90% (Table 2). Compared to MRTX849-KRAS G12C, it was observed that the frequency of hydrophobic interactions with the inhibitor were greatly reduced by 10.32%, 60.90%, 4.78%, and 11.26% for the residues S68, T16, C96 and D96 in double mutant KRAS. Notably, the frequency of π–π interaction between residues Y96 and MRTX849 was 38.52% in KRAS G12C. However, Both the hydroegn bond interaction and π–π interaction frequency completely disappeared after K16T and Y96C/D mutation (Figure 11B). From the above analysis, it is suggested that the second-site mutation in protein-inhibitor interaction could reduce the binding affinity of inhibitor to KRAS, resulting in the occurrence of drug resistance.

### 2.8. Dynamics Network Analysis

To investigate how the second-site mutation at the switch-Ⅱ binding site affects the network connectivity of the KRAS protein. In this study, all snapshots were extracted from the MD trajectory for each system to conduct dynamical network analysis. As shown in Figure 12, each color denotes a different community, for the five Apo forms mutated KRAS (G12C, G12C-R68S, G12C-K16T, G12C-Y96C and G12C-Y96D), the system is divided into 5, 12, 11, 15, 7 communities, respectively. And in the AMG510 (MRTX849) bound complexes, 9 (5), 8 (7), 11 (11), 12 (11), 10 (5) communities in KRAS G12C, G12C-R68S, G12C-K16T, G12C-Y96C and G12C-Y96D, respectively.

According to per-residues energy decomposition results, the residues in protein and inhibitor located in switch-Ⅱ of KRAS are used as “source” and “sink” nodes for suboptimal pathways analysis. Each residue in the KRAS protein represents a node, and the edge (or connection) between two nodes represents the total interaction between two residues. As shown in Figure 13. For the AMG510-bound complexes, the count of allosteric signal pathways involves node (C12, V14, M72, T58 and K11 of G12C-R68S; C12, V14, M72, L53 and I55 of G12C-K16T; M72, Mg^2+^, I53, I55 and D69 of G12C-Y96C; R68, Y71, K5, D54 and K16 of G12C-Y96D) are increased, and node (I21,A18, Mg^2+^, GDP and K16 of G12C-R68S; I21, D33, I24, V29 and T16 of G12C-K16T; I24, I21, A18, V29 and K16 of G12C-Y96C; Y32, Mg^2+^, V29, I21 and T20 of G12C-Y96D) are decreased compared to AMG510-KRAS G12C (Figure 13A). In particular, except for G12C-Y96D, the residue at position 16 show a decrease in allosteric signal connections relative to KRAS G12C (Figure 13A). For the MRTX849 bound complexes, the count of allosteric signal involves node (V9, K5, V7, Mg^2+^ and D54 of G12C-R68S; A59, D57, K5, M72 and G75 of G12C-K16T; A11, V9, G13, V7, D69 of G12C-Y96C; V8, L56, Mg^2+^, T20 and S39 of G12C-Y96D) are increased, and s (E31, C12, T58, A18 and K16 of G12C-R68S; A18, V14, Mg, T16 and C12 of G12C-K16T; GDP, Mg^2+^, A18, K16, C12 of G12C-Y96C; K16, T58, E31, V29 and A18 of G12C-Y96D) are decreased with respect to MRTX849-KRAS G12C (Figure 13B). In addition, it was observed that both residues at position 16 and 18 show a decrease in allosteric signal connections relative to KRAS G12C during MD simulation. The dynamics network analysis results indicated that variations in allosteric relationship between inhibitor and second-site mutations at position (residues 61, 68 and 96) evidently alters conformations of the switch domain in KRAS. These differences may affect the activity of KRAS inhibitors.

## 3. Materials and Methods

### 3.1. Construction of Simulated Systems

The initial crystal structure for AMG510 and MRTX849 in complex with KRAS G12C was obtained from the Protein Data Bank database (PDB ID code:6OIM and 6UT0) [28,29]. In this study, the missing residues on AMG510-KRAS G12C (PDB ID code:6OIM) were added using *Chimera v1.15rc* [34]. As the crystal structure of second-site mutant are not available from the PDB database, the 3D structure of the four mutants were constructed using the *PyMOL v1.3* mutagenesis tool [35] to mutate the lysine, arginine, tyrosine located in position 16, 68, 96 and 96 of the selected two crystal structures back to threonine, arginine, cysteine and aspartic acid, respectively. Then, five apo forms of the 3D structures of G12C, G12C-R68S, G12C-Y96C and G12C-Y96D were generated by deleting the inhibitor from the switch-II pocket of KRAS protein. The *Protein Prepare Wizard module* [36] in Maestro was used for the preparation and the minimization of energy of all protein structures. The program *LEaP* embedded in AMBER18 [37] with the ff14SB force field used for the protein, including adding all missing hydrogen atoms of the protein. AMBER force field (gaff) parameter sets for the covalent inhibitors (AMG510 and MRTX849) and the allosteric cysteine. The partial charges were derived from RESP calculation using an HF/6-31G* electrostatic potential calculated by Gaussian09 [38]. Additionally, the force field parameters of GDP and magnesium ions were obtained from the AMBER Parameter Database [39] (http://research.bmh.manchester.ac.uk/bryce/amber, accessed on 7 September 2020). These ten systems were placed in a cubic water box containing 0.15 M NaCl aqueous solution and the minimum distance between the solute and the box boundary was 12 Å. Water was described with the TIP3P water model.

### 3.2. Molecular Dynamics Simulation

GPU-accelerated AMBER18 was used to perform all MD simulations. 2000-step steepest descent minimization, 1000-step conjugate gradient minimization, 2 ns heating process of 0 to 310 K at the NVT condition and a 2 ns equilibrium process of 310 K at the NPT condition were sequentially run to ten independent 500 ns on each simulated system. Meanwhile, the electrostatic interactions were calculated using the Particle Mesh Ewald method with a cutoff of 8.0 Å for long-range interactions [40]. The covalent bonds containing hydrogen atoms were constrained by the Shake algorithm [41]. The simulated step size was 2 fs. Periodic boundary conditions were used in the system. The simulation frames were outputted once every 100 ps.

### 3.3. Thermodynamics Analysis

To investigate the thermodynamic properties of KRAS protein−inhibitors interactions, based on the system stability determined by the fluctuation of root-mean-square deviations (RMSDs), 5000 snapshots were extracted from the corresponding ten independent MD trajectories. The binding free energy for the inhibitors AMG510 and MRTX849 to KRAS was calculated using the MM/GBSA method [42,43,44] according to the following equation:$$\Delta G_{\mathrm{cal}}=\Delta E_{\mathrm{ele}}+\Delta E_{\mathrm{vdW}}+\Delta E_{\mathrm{covalent}}+\Delta G_{\mathrm{solv}}-T\Delta S=\Delta E_{\mathrm{bing}}-T\Delta S,$$
where the first three terms (ΔE_ele_, ΔE_vdW_ and ΔE_covalent_) represent the electrostatic, van der Waals and covalent energy contributions, respectively; The four term (ΔG_solv_) is the polar and nonpolar contributions to solvation free energy; and −TΔS is the conformational entropy change. The electrostatic (ΔE_ele_) and van der Waals (ΔE_vdW_) contributions are computed using the molecular mechanics energy function. The nonpolar contributions (ΔG_solv_) were estimated by the solvent-accessible surface area (SASA) determined using a water probe radius of 1.4 Å and the surface tension constant c was set to 0.0072 kcal/(mol/Å^2^) [45]. The contribution of entropy changes (−TΔS) were estimated by normal mode analysis [46] Due to the high computational cost in the entropy calculation, 50 snapshots from each MD trajectory were used for calculation.

### 3.4. Construction and Verification of Markov State Models

Markov state models (MSMs) were constructed to predict the long time scale dynamics of the mutated KRAS. In this study, two switch regions of 3D coordinate atoms in protein are selected as the metric of MSMs when the RMSF values are greater than >2 Å. First, clustering of the trajectories was performed using the k-means clustering to partition the MD trajectory snapshots in 5000 clusters by the R-3.6.3 software package [47]. Then, the number of cluster representatives at the mean RMSD <1 Å were chosen as the disperse MD trajectory for MSMs. After dimensionality reduction and clustering, the microstates were merged into metastates. Each metastate consists of 10 representative conformations. In these conformations, due to the main fluctuation occurring in the two loop regions of the KRAS protein, the mean RMSD of the Loop Ⅰ and Loop Ⅱ of KRAS protein with respect to its initial conformation is defined as the RMSD of the corresponding metastate. After MSM analysis, the conformational state with the lowest RMSD is determined as the initial state, and the conformational state with the highest proportion is set as the end state. The transition path theory (TPT) [48,49] was used to calculate the fluxes of these pathways and the pathways from each state to the end state are identified. The flux of a pathway can be understood as the probability that the reaches the end state along that pathway. The construction and verification of MSMs were carried out by the PyEMMA-2.5.7. Python packages [50].

### 3.5. Potential of Mean Force Analysis

The potential of mean force (PMF) based on the MD simulations have been applied to explore the detailed free energy landscapes for the conformational transition processes of mutated KRAS. Boltzmann’s equation was used to calculate the PMF profiles and to detect the potential distributions via the simulation potential frames [51]. The RMSD of loop Ⅰ (residues 30–42) and loop Ⅱ regions (residues 57–63) of KRAS were chosen as reaction coordinate to construct two-dimensional (2D) PMF profiles on each simulated system.

### 3.6. Interaction Fingerprints Analysis

In order to identify which residues are involved in a specific type of interaction quantitatively, 5000 frames were evenly extracted from the 5 μs MD trajectory, and the protein-ligand interaction fingerprints were performed using *ProLIF* [52]. For each frame, the calculation zones of the protein-ligand complex were defined as the regions within 6 Å of the ligand. During the calculation, fourteen interaction types (including Hydrophobic, HBAcceptor, HBDonor, PiStacking, etc.) of the atoms of proteins and ligands were evaluated.

### 3.7. Dynamic Network Analysis

A dynamical network analysis was performed to identify possible allosteric signal pathways in which perturbations propagate throughout MD trajectories. The analysis was done using NetworkView plugin in *VMD v1.93* software [53,54]. Each Cα atom in protein, heavy atoms in ligand and metal ions were chosen as a communicating node. If the edge distances between nodes remain within 4.5 Å during the 75% of the MD simulation time, then there was an edge connecting the two nodes as rods representation. The edge distance between two nodes formed the pathways. The edges were weighted using a pairwise correlation coefficient matrix and analyzed by the Carma program. Based on this correlation matrix, the simulated systems were divided into many sub networks called communities. By defining the “source” and “target” residues, those allosteric connections are constructed by the *subopt* program, and the shortest allosteric path between them is termed the optimal path. All others were defined as suboptimal paths.

### 3.8. Pharmacophore Modeling

The 3D pharmacophore model of KRAS G12C inhibitor with mutant protein was built on the module of Schrodinger Maestro [55]. First, the ten representative structures were obtained from the main free energy basins. Second, the protein−ligand workspace pharmacophore of each complex was utilized to generate key chemical features by manually inspecting the interaction, whose features are important for binding. The receptor-based excluded volume with a scaling factor of 0.50 for van der Waals atom radii was then added. Finally, the pharmacophore of each inhibitor-binding complex generated the chemical features with the hydrogen bonds, hydrophobic interactions, and the excluded volume shell.

## 4. Conclusions

In this work, long time MD simulation, binding free energy calculation, MSMs, 2D-free energy landscape, molecular interaction fingerprint and dynamic network analysis were employed to investigate the structural and energetic features of KRAS G12C and its four drug resistant variants to AMG510 and MRTX849. The detailed binding free energies calculated by MM/GBSA and protein-ligand interaction fingerprint analysis suggest that the second-site mutation directly induces obvious drug resistance toward AMG510 and MRTX849. The construction of MSMs and 2D-free energy landscape implied that the second-site mutation on protein affects the conformational transformation of the switch domains between the closed and open conformation compared to inhibitor-KRAS G12C, especially from the switch-Ⅰ domain may affect the binding of downstream effector protein. The dynamic network analysis shows that allosteric apo or mutated inhibitor-KRAS exert different allosteric regulations on the two switch regions of the protein. Our study shows that a molecular level explanation for drug resistance to the results will be helpful for the design of novel inhibitors with improved potency for the treatment of cancer.

## Figures and Tables

**Figure 1 ijms-23-13845-f001:**
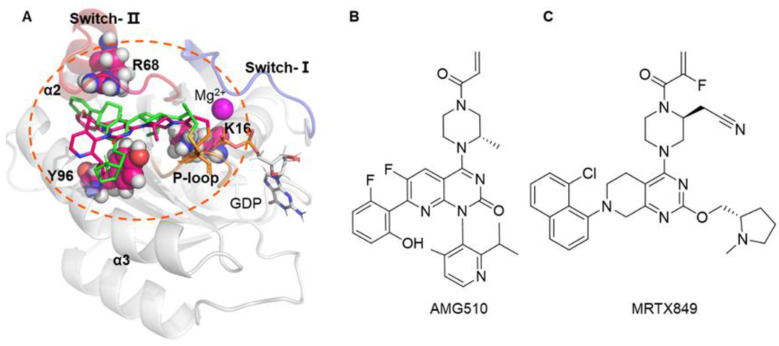
(**A**) Overall structure of the KRAS G12C complex with AMG510 and MRTX849. The protein and ligand are shown as cartoons and stick, respectively. The mutated residues in switch-Ⅱ binding pocket are shown as sphere. (**B**,**C**) Chemical structure of AMG510 and MRTX849.

**Figure 2 ijms-23-13845-f002:**
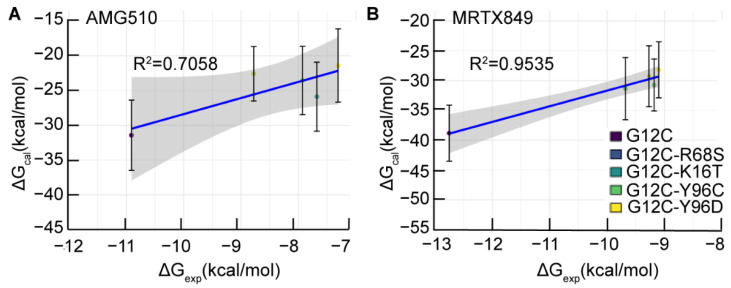
The correlation coefficient between experiment values (ΔG_exp_) and the predicted binding free energies (ΔG_cal_) obtained with the MM/GBSA method for (**A**) AMG510-bound complexes and (**B**) MRTX849-bound complexes. The gray shaded regions represent the confidence interval.

**Figure 3 ijms-23-13845-f003:**
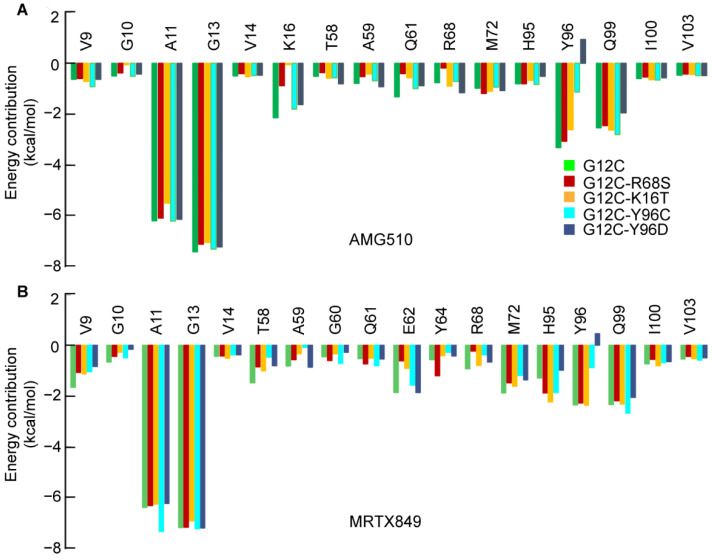
Per-residue energy profiles for the binding of AMG510 (**A**) and MRTX849 (**B**) binding to KRAS G12C, G12C-R68S, G12C-K16T, G12C-Y96C and G12C-Y96D, respectively.

**Figure 4 ijms-23-13845-f004:**
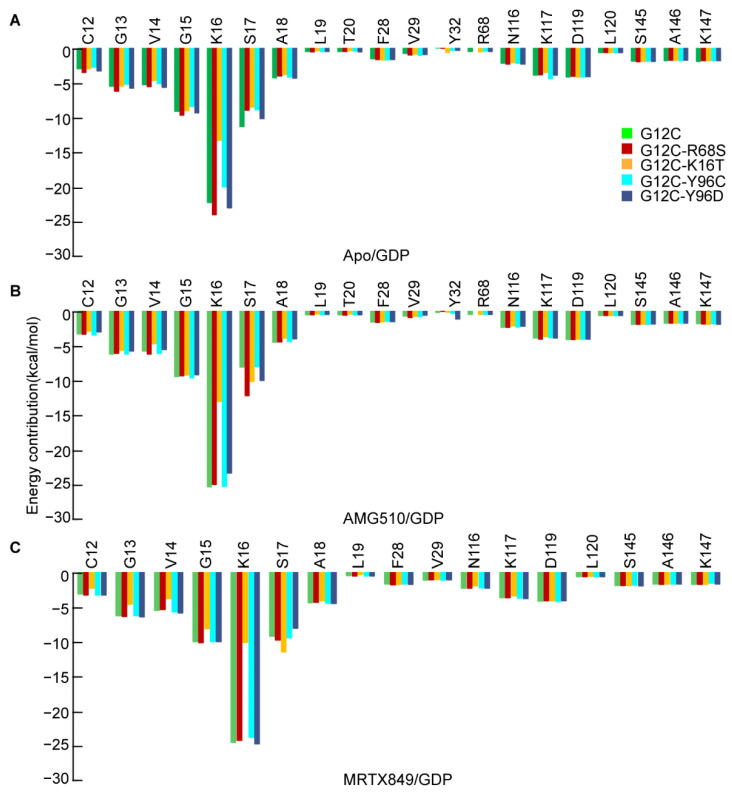
Per-residue energy profiles for GDP binding in apo (**A**), AMG510 (**B**) and MRTX849 (**C**) inhibitor bound to mutated KRAS G12C, G12C-K16T, G12C-R68S, G12C-Y96C and G12C-Y96D, respectively.

**Figure 5 ijms-23-13845-f005:**
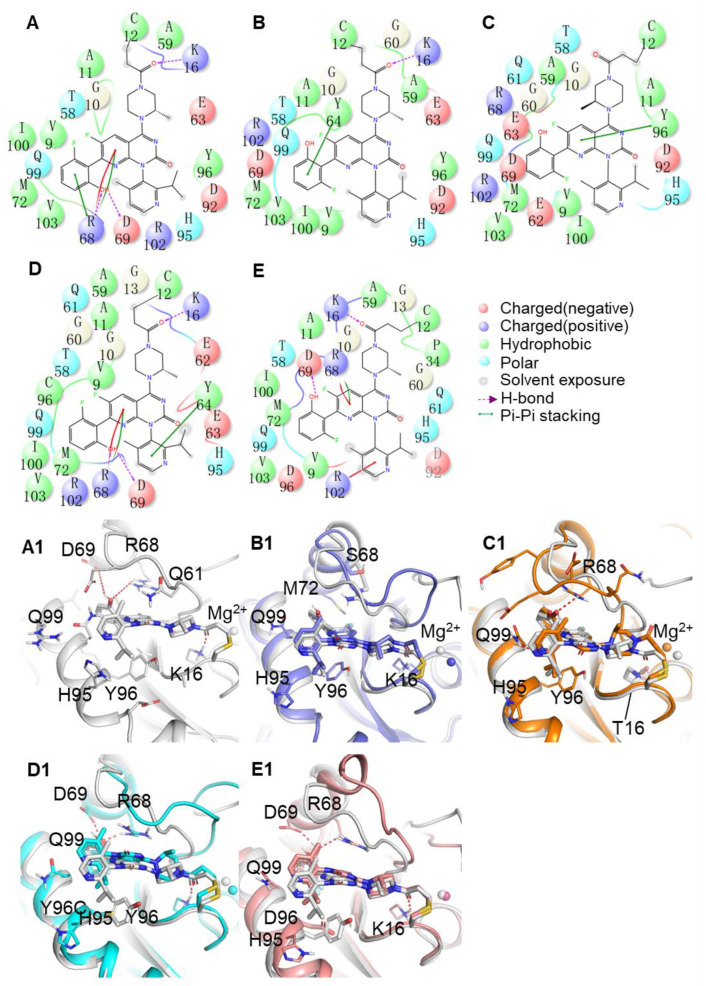
(**A**–**E**) Binding modes of the studied AMG510 bound to KRAS G12C, G12C-R68S, G12C-K16T, G12C-Y96C and G12C-Y96D, respectively. (**A1**–**E1**) The 3D presentations of the binding modes in the switch-Ⅱ pocket of KRAS G12C, G12C-R68S, G12C-K16T, G12C-Y96C and G12C-Y96D with AMG510 shown in gray, orange, blue, cyan, and pink, respectively. The Mg^2+^ is shown as a sphere.

**Figure 6 ijms-23-13845-f006:**
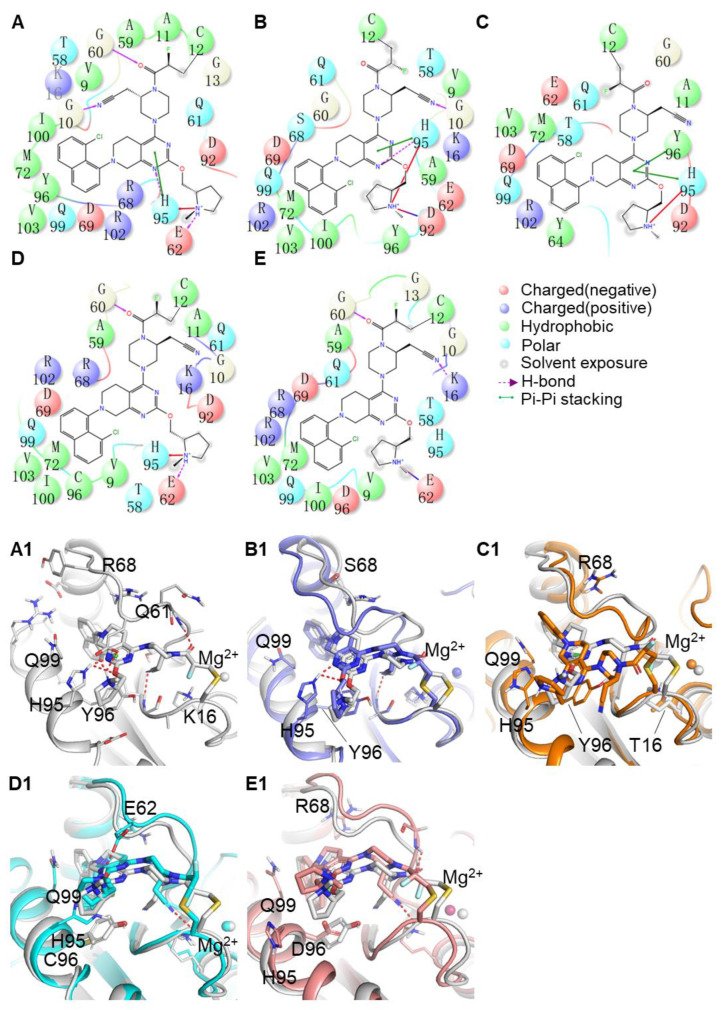
(**A**–**E**) Binding interactions in 2D representation of the studied MRTX849 bound to KRAS G12C, G12C-R68S, G12C-K16T, G12C-Y96C and G12C-Y96D, respectively. (**A1**–**E1**) The 3D presentations of the binding modes in the switch-Ⅱ pocket of KRAS G12C, R68S, G12C-K16T, G12C-Y96C and G12C-Y96D with AMG510 shown in gray, orange, blue, cyan, and pink, respectively. The Mg^2+^ is shown as a sphere.

**Figure 7 ijms-23-13845-f007:**
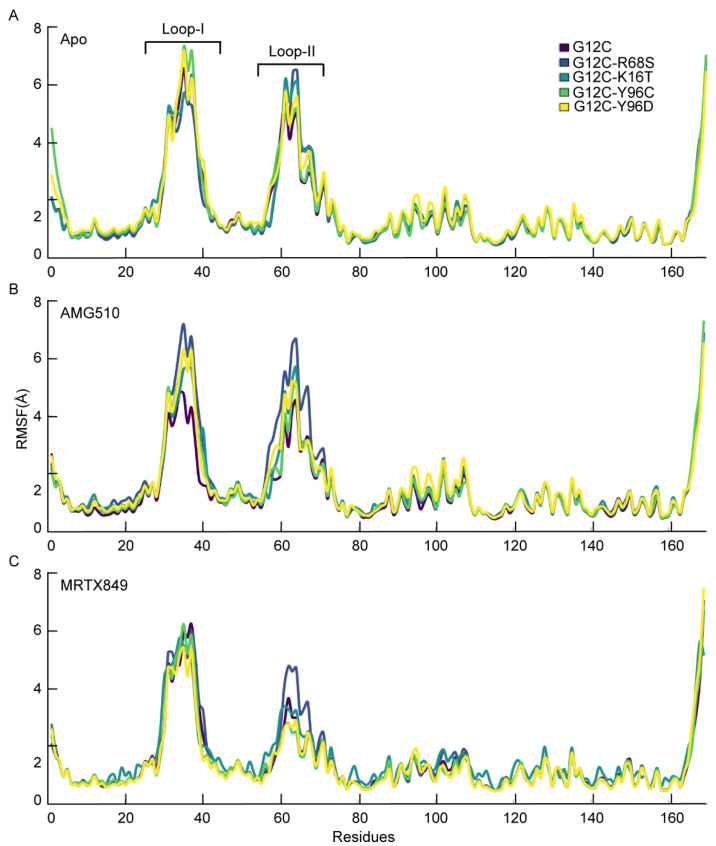
(**A**) The RMSF values (Å) of residues in apo KRAS G12C and four mutated systems. (**B**,**C**) The backbone RMSF values (Å) of residues in AMG510/MRTX849 bound G12C complexes and AMG510/MRTX849 bound mutant complexes.

**Figure 8 ijms-23-13845-f008:**
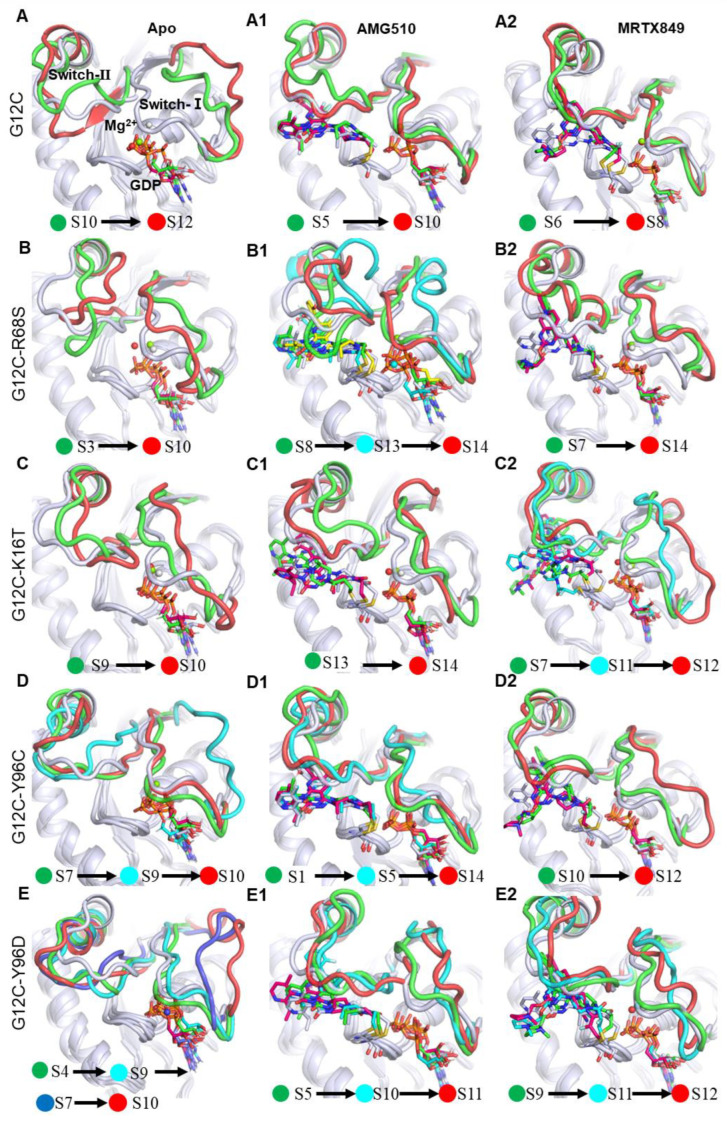
The main flux metastable state (Si) with one representative structure were identified for each system, and the initial crystal structure of KRAS G12C is shown in gray color. The green and cyan (or yellow) label denotes the start and end state. (**A**–**E**) The apo form KRAS G12C and four second-site mutilation KRAS. The AMG510 (**A1**–**E1**) and MRTX849 (**A2**–**E2**) bound to KRAS G12C and four second-site mutation KRAS. The metastable state (Si) conformations with a total of probability flux >60.0% are depicted by cartoon.

**Figure 9 ijms-23-13845-f009:**
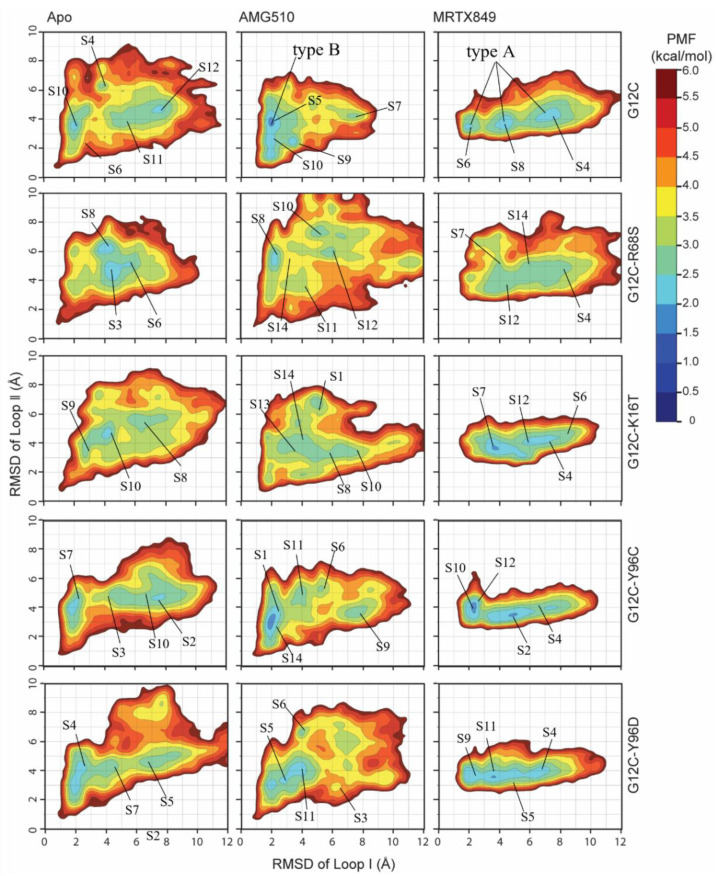
2D potential of mean force (PMF) profiles of protein Loop Ⅰ and Loop Ⅱ of RMSDs as reaction coordinates. The first column displays the corresponding apo form KRAS G12C and four second-site mutilation KRAS. The second column pictures the AMG510 bound to KRAS G12C and four second-site mutilation KRAS. The third column depicts the MRTX849 bound to KRAS G12C and four second-site mutilation KRAS.

**Figure 10 ijms-23-13845-f010:**
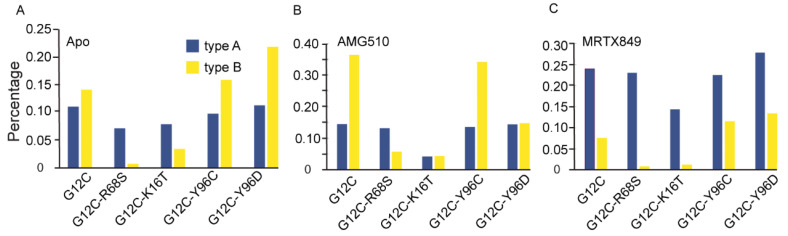
The percentage of two conformational type A and type B in five apo (**A**), five AMG510-bound (**B**), and five MRTX849-bound (**C**) complexes, respectively.

**Figure 11 ijms-23-13845-f011:**
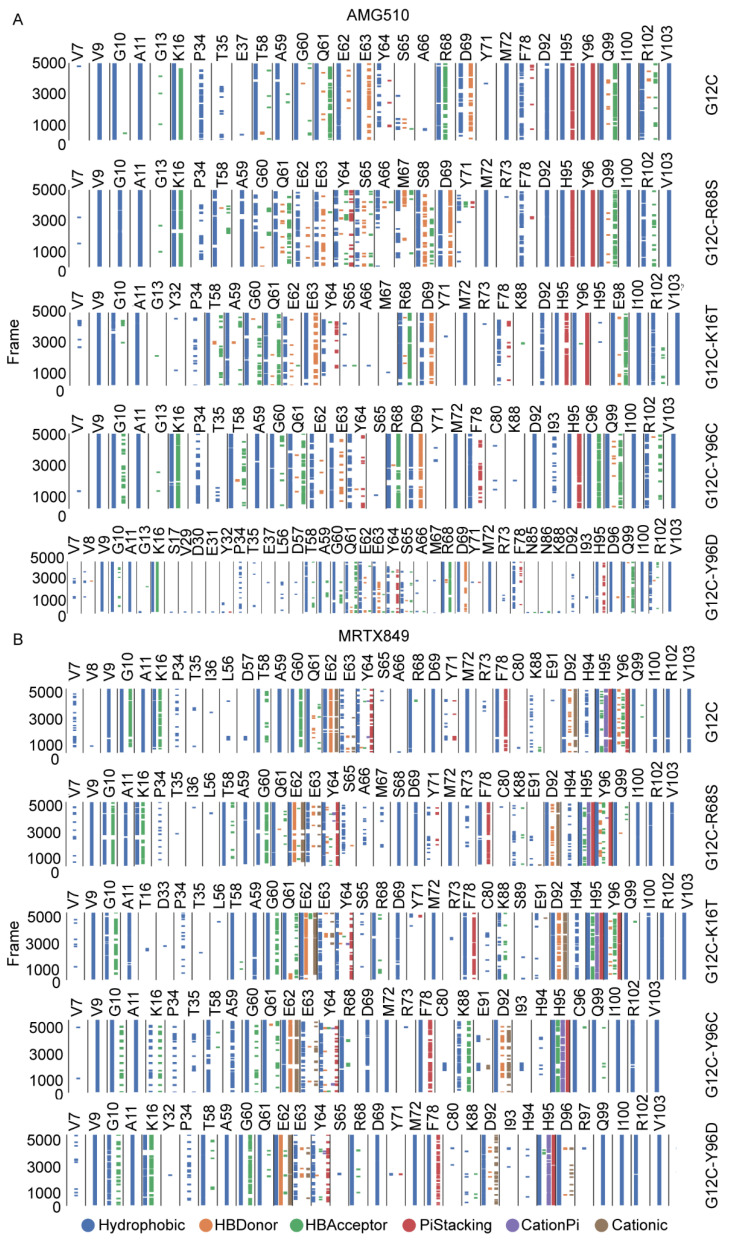
Interaction fingerprints of AMG510 (**A**) and MRTX849 (**B**) with mutated KRAS during MD simulation. Each of the interaction types is colored by different color.

**Figure 12 ijms-23-13845-f012:**
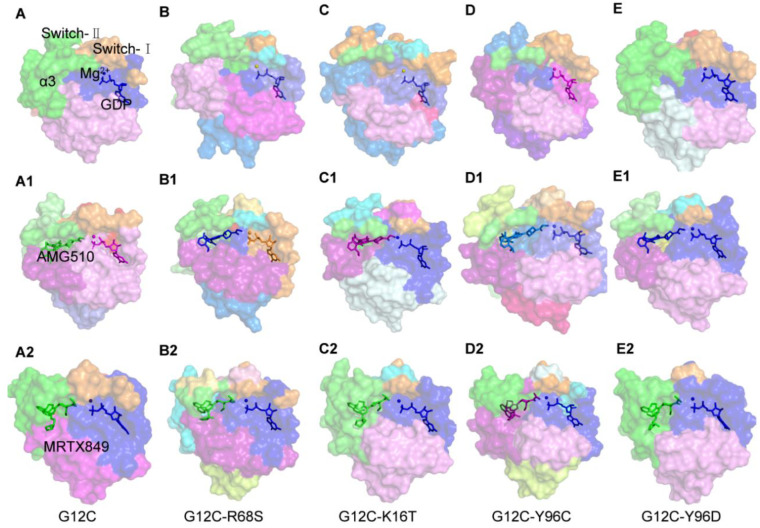
Communities distribution analysis for each simulated system. (**A**–**E**) Dynamical network analysis of apo form KRAS G12C, G12C-R68S, G12C-K16T, G12C-Y96C and G12C-Y96D. Dynamical network with AMG510 (**A1**–**E1**) and MRTX849 (**A2**–**E2**) of KRAS G12C, G12C-R68S, G12C-K15T, G12C-Y96C and G12C-Y96D protein, respectively. Each color denotes a different community.

**Figure 13 ijms-23-13845-f013:**
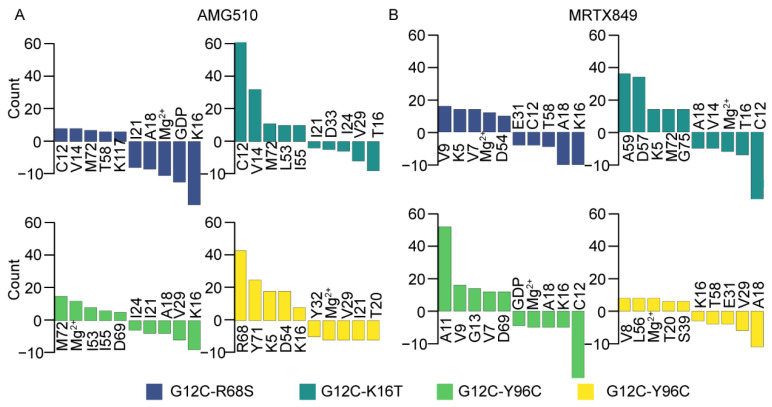
Allosteric signal statistical analysis forAMG510 (**A**) and MRTX-bound (**B**) complexes. A positive value indicates an increase in the number of signal pathways compared to the corresponding KRAS-G12C system. A negative value indicates a decrease in the number of signal pathways compared to the corresponding KRAS-G12C system. Residues located at the two switch regions and the C1′ atom on the GDP are used for allosteric connectivity analysis.

**Table 1 ijms-23-13845-t001:** Binding affinities of the AMG510 mutant complexes from calculation and experiment.

Energy Components (kcal/mol)	AMG510-Bound System				
	G12C	G12C-R68S	G12C-K16T	G12C-Y96C	G12C-Y96D
ΔE_vdW_	−62.48 ± 4.92	−51.83 ± 6.46	−58.18 ± 7.21	−56.68 ± 6.67	−52.55 ± 6.45
ΔE_elec_	−152.11 ± 7.46	−147.85 ± 10.66	−146.96 ± 10.80	−148.16 ± 9.23	−145.39 ± 10.94
ΔE_covalent_	89.06 ± 1.51	89.12 ± 1.50	89.83 ± 1.42	88.94 ± 1.51	89.32 ± 1.48
ΔG_solv_	59.13 ± 6.30	52.20 ± 9.51	55.93 ± 8.12	54.48 ± 7.36	52.01 ± 8.88
^a^ ΔE_bind_	−67.39 ± 5.15	−58.36 ± 6.97	−59.38 ± 6.52	−61.42 ± 6.8	−56.59 ± 6.11
^b^−TΔS	35.96 ± 5.51	35.78 ± 3.92	35.84 ± 4.91	35.54 ± 4.96	35.19 ± 5.27
^c^ ΔG_cal_	−31.43	−22.58	−23.54	−25.88	−21.40
^d^ ΔΔG_cal_		8.85	7.89	5.55	10.03
^e^ ΔG_exp_	−10.91	−8.78	−7.93	−7.68	−7.31
IC_50_(nM)	19.81	643.27	2524	3814	6920.67

All energies are in kcal/mol, with corresponding standard errors. The covalent energy terms (ΔG_covalent_) as well as the 1–4 electrostatic (E_elec_) and van der Waals (E_vdW_) terms were all included. ^a^ Calculated binding free energy using MM/GBSA method. ^b^ The entropy calculated through normal mode analysis. ^c^ ΔG_cal_ = ΔE_bind_ − TΔS; ^d^ ΔΔG_cal_ = ΔG_double-mutant_ − ΔG_G12C_; ^e^ Experimental binding free energy calculated using ΔG_exp_ = RTln(IC_50_). ^c^ IC_50_ values were from the experimental literature [15].

**Table 2 ijms-23-13845-t002:** Binding affinities of the MRTX849 mutant complexes from calculation and experiment.

Energy Components (kcal/mol)	MRTX849-Bound System				
	G12C	G12C-R68S	G12C-K16T	G12C-Y96C	G12C-Y96D
ΔE_vdW_	−71.53 ± 3.94	−62.38 ± 8.77	−63.40 ± 6.77	−62.05 ± 4.79	−56.34 ± 9.84
ΔE_elec_	−306.89 ± 29.32	−299.89 ± 28.64	−319.38 ± 29.57	−281.80 ± 21.47	−308.99 ± 22.98
ΔE_covalent_	89.99 ± 1.48	89.71 ± 1.44	90.35 ± 1.56	89.67 ± 1.42	89.64 ± 1.49
ΔG_solv_	211.95 ± 27.82	204.29 ± 28.15	226.33 ± 28.39	187.00 ± 20.53	215.42 ± 22.71
^a^ ΔE_bind_	−76.48 ± 7.28	−68.28 ± 7.70	−66.10 ± 8.22	−67.18 ± 5.27	−60.39 ± 8.62
^b^−TΔS	37.67 ± 4.70	36.94 ± 5.22	36.84 ± 5.11	36.45 ± 4.36	34.81 ± 4.78
^c^ ΔG_cal_	−38.81	−31.34	−29.26	−30.73	−28.18
^d^ ΔΔG_cal_		7.47	9.55	8.08	10.63
^e^ ΔG_exp_	−12.75	−9.68	−9.27	−9.18	−9.10
IC_50_(nM)	1.01	147.07	289.47	332.90	376.64

All energies are in kcal/mol, with corresponding standard errors. The covalent energy terms (ΔG_covalent_) as well as the 1–4 electrostatic (E_elec_) and van der Waals (E_vdW_) terms were all included. ^a^ Calculated binding free energy using MM/GBSA method. ^b^ The entropy calculated through normal mode analysis. ^c^ ΔG_cal_ = ΔE_bind_ − TΔS; ^d^ ΔΔG_cal_ = ΔG_double-mutant_ − ΔG_G12C_; ^e^ Experimental binding free energy calculated using ΔG_exp_ = RTln(IC_50_). ^c^ IC_50_ values were from the experimental literature [15].

**Table 3 ijms-23-13845-t003:** Fingerprints of molecular interaction including the interaction types between AMG510 and mutant KRAS protein.

Residues	Interaction Type	Frequency (%)				
		G12C	G12-R68S	G12C-K16T	G12C-Y96C	G12C-Y96D
V9	Hydrophobic	93.78	77.52	77.42	95.44	68.04
G10	Hydrophobic	67.94	64.06	0.00	68.78	32.80
A11	Hydrophobic	72.70	73.30	39.42	79.40	56.64
16	Hydrophobic	77.04	73.72	0.00	68.30	54.02
	HBAcceptor	85.84	84.16	0.00	78.74	69.06
T58	Hydrophobic	84.54	71.74	80.60	84.74	67.60
A59	Hydrophobic	83.12	70.82	56.18	78.66	77.70
G60	Hydrophobic	72.16	37.82	57.28	64.00	60.34
Q61	Hydrophobic	78.12	40.56	54.40	62.24	55.86
E62	Hydrophobic	33.34	0.00	0.00	0.00	0.00
E63	Hydrophobic	56.62	33.72	57.86	43.26	51.84
68	Hydrophobic	61.20	0.00	76.62	64.30	84.44
M72	Hydrophobic	99.56	99.02	98.84	98.92	95.28
H95	Hydrophobic	99.66	99.36	97.86	97.52	60.00
96	Hydrophobic	100.00	100.00	99.96	98.44	85.82
Q99	Hydrophobic	99.98	99.90	99.96	99.94	95.22
I100	Hydrophobic	43.96	43.20	62.14	76.82	70.48
V103	Hydrophobic	86.10	81.74	84.32	84.30	70.14

**Table 4 ijms-23-13845-t004:** Fingerprints of molecular interaction including the interaction types between MRTX849 and mutant KRAS protein.

Residues	Interaction Type	Frequency (%)				
		G12C	G12C-R68S	G12C-K16T	G12C-Y96C	G12C-Y96D
V9	Hydrophobic	89.20	83.46	90.76	98.82	93.42
G10	Hydrophobic	75.92	72.84	52.52	43.82	0.00
	HBAcceptor	48.92	54.86	0.00	0.00	0.00
A11	Hydrophobic	32.34	0.00	41.06	76.36	41.62
16	Hydrophobic	60.90	50.58	0.00	0.00	41.76
T58	Hydrophobic	91.06	84.84	74.02	40.08	80.42
A59	Hydrophobic	71.84	78.16	56.96	32.46	77.54
G60	Hydrophobic	87.62	77.00	79.66	89.52	86.02
Q61	Hydrophobic	87.02	83.64	84.38	88.62	90.98
E62	Hydrophobic	82.60	80.66	80.72	86.26	97.26
	HBDonor	45.70	0.00	37.46	0.00	33.48
	Cationic	44.36	0.00	36.42	0.00	32.46
Y64	Hydrophobic	0.00	60.16	32.82	0.00	0.00
	PiStacking	0.00	35.76	0.00	0.00	0.00
68	Hydrophobic	84.12	52.88	84.32	64.46	87.56
D69	Hydrophobic	36.64	59.32	0.00	0.00	0.00
M72	Hydrophobic	99.60	99.12	99.78	99.64	98.42
F78	Hydrophobic	43.60	39.94	42.22	0.00	0.00
D92	Hydrophobic	59.72	59.96	66.50	61.32	0.00
H95	Hydrophobic	90.84	94.90	91.56	93.08	74.84
	PiStacking	0.00	0.00	44.22	52.08	0.00
96	Hydrophobic	98.08	99.26	99.66	93.30	86.82
	PiStacking	38.52	33.04	50.26	0.00	0.00
Q99	Hydrophobic	95.84	98.80	99.92	100.00	99.44
I100	Hydrophobic	83.80	80.96	82.80	90.06	87.86
V103	Hydrophobic	86.48	86.82	88.26	82.16	86.06

## Data Availability

The raw data supporting the conclusions of this article will be made available by the authors, without undue reservation.

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
