# Peer review of "In Silico Study of the Acquired Resistance Caused by the Secondary Mutations of KRAS G12C Protein Using Long Time Molecular Dynamics Simulation and Markov State Model Analysis"

_ijms, 2022, doi:10.3390/ijms232213845_

Round 1
Reviewer 1 Report
Tu et al present a detailed analysis of KRAS G12 variants and its inhibitors using several bioinformatics tools/approaches. The is a well scientific and quite detailed analysis. It would be suitable for publication after addressing the following questions/comments:
1. Bars in figures 3 and 4 need a different coloring scheme for easy visualization.
2. As per my understanding a total of 15 simulations were conducted and none of them were done in duplicates. What do the authors think about the significance of smaller differences in their calculations for fluctuations, free energy calculations?
3. Since the authors has already performed a rigorous analysis of the protein-ligand interactions describing the effects of mutations to changes in binding affinities. Could they also suggest an ideal pharmacophore that would bind effectively to the protein.
Author Response
Yao-Response.to.Reviewer.1

Reviewer 2 Report
KRAS is a small GTPase protein that is vital to KRAS mutant cancers. There are two inhibitors approved by FDA that target KRAS G12C. However, the second-site mutation in KRAS brings drug resistance and the mechanism remains unclear. In this study, the authors conducted long MD simulations and apply a series of computational tools, ranging from Markov state model, free energy calculation, potential of mean force analysis, interaction fingerprints analysis and dynamic network analysis, to elucidate the mechanism. Key conformations and residues are identified that might be helpful for the design of novel inhibitors. Overall, this work is well presented and is worth publication with the following points being addressed.
Major:
1. One of the common MSM workflows is dimensionality reduction + MSM construction, see [The Journal of Physical Chemistry B, 124(41), 8960-8972]. In this study, the authors selected loop I and II regions due to high fluctuation. While this is reasonable, the authors should mention and discuss the usual situation.
Minor:
1. Figure 8 is with very low resolution. The authors should replace this with clear plots.
2. The sentence in line 65 is incomplete.
3. MSM abbreviation should be noted in the first appearance in line 61-62.
4. Line 171, one space is missing for “forMRTX849”.
Author Response
Yao-Response.to.Reviewer.2

Round 2
Reviewer 2 Report
The authors have addressed all my concerns. The work is now worth publication.